# Effect of Heating Process on Microstructure and Properties of 2205/Q235B Composite Interface

**Fengqiang Xiao** [1,2,3], **Dongpo Wang** [1,2,*], **Zhiming Gao** [1] and **Lanju Zhou** [4]

[1] School of Materials Science and Engineering, Tianjin University, Tianjin 300350, China; junweiwy@163.com or xiaofengqiang@luxin.cn (F.X.); gaozhiming@tju.edu.cn (Z.G.)
[2] Tianjin Key Laboratory of Advanced Joining Technology, Tianjin Univesity, Tianjin 300350, China
[3] Shandong High-Tech Investment Corporation, Jinan 250101, China
[4] Shandong Iron & Steel Group Co., LTD, Jinan250101, China; zhoulanju@sdsteelrz.com
[*] Correspondence: wangdp@tju.edu.cn

**Abstract:** In this paper, the influence of heating process parameters on interface characteristics and mechanical properties of 2205/Q235B clad steel plate was systematically studied. The results showed that the interfacial gap of the 2205/Q235B composite blank was completely bonded by the mutual diffusion of elements under the action of temperature and metallurgical bonding is achieved. The shear strength of the air-cooled samples was only 114–132 MPa, which was far lower than that of water-quenched samples and rolling deformation samples and was unable to meet the requirements of engineering applications. With the increase in heating temperature and holding time, the diffusion distance of the Cr element gradually increased. After rolling deformation, the diffusion distance of the Cr element was significantly reduced to 4.1–10.2 μm. Rolling deformation of the specimen in the decarburization showed the lowest microhardness, and in combination with the microhardness of the interface is about 236–256 HV, which is between the hardness of Q235B and 2205. The 2205 stainless-steel shows the lowest corrosion rate and the best corrosion resistance when rolling at 1200 °C. It was found that the corrosion was the most significant in the side of Q235B near the bonding zone. The corrosion pit width increased gradually with increased heating temperature.

**Keywords:** heating temperature; heat-up time; 2205/Q235B; interface; shear strength

## 1. Introduction

Metal composite plate is a composite material formed by metallurgical bonding of metals with different properties on the interface by various cladding technologies, such as explosive composite [1–5], rolling composite [6–8], or explosive and rolling composite [9]. Metal composite plate has both the strength characteristics of the base material and the corrosion resistance, heat resistance, and other functional properties of the cladding material. It not only meets the special requirements of different environments and use conditions but also saves on precious alloy elements, reducing the overall material cost [10–13]. Due to its obvious performance and cost advantages, metal composite materials have been widely used in the petrochemical industry, marine engineering, electric power, and other fields and industries [14–16].

In the research of metal composite plates, most scholars have systematically studied and discussed the microstructure and mechanical properties of the composite plate from two aspects: Rolling technology and heat treatment technology. Dhib Z. et al. [17] investigated the correlation between microstructure and mechanical properties in low-carbon steel/austenitic stainless-steel clad composite fabricated by hot-roll bonding. Wang et al. [18] found that the refinement degree of microstructure was increased with the increasing rolling reduction ratio. The thicknesses of the decarburized,

carburized layers, and martensite zone were decreased. Moreover, the interface bonding strength, tensile strength, and interface deformation coordination were increased. Yu et al. [19] fabricated a 304 stainless-steel/medium carbon steel clad plate by hot-rolling in an argon atmosphere. The rolling parameters played an important role on the shear strength and composition diffusion.

To better meet the requirements of metal composite plates in specific application fields and further improve the comprehensive properties of these materials, researchers have conducted considerable research on heat treatment technology. Song et al. [20] studied the influence of normalizing treatment, quenching, and tempering treatment on the structure and properties of S32750/EH40 composite plate. The samples quenched at 1080 °C for 1 h and then tempered at 580 °C for 1–2 h showed better mechanical properties than those treated with normalizing, and the yield strength and tensile strength increased with the increase of heat treatment temperature. Wang et al. [21] took 316L/Q370qE as the research object and carried out heat treatment at the four temperatures of 500 °C, 600 °C, 800 °C, and 1000 °C, respectively. The results show that the diffusion region of Ni, Cr, and other elements becomes wider with the increase of heat treatment temperature, and the shear strength of the composite plates also increases. Jin et al. [22] improved the mechanical properties and corrosion resistance of 304/Q345R stainless-steel-clad plate through changing the heat treatment processes. After being oil cooled to 450 °C and air cooled, the strength, plasticity as well as corrosion resistance of the clad plate were significantly enhanced. Kosturek et al. [23] used the explosive welding to bond the Inconel 625 and P355NH steel. The microstructure of the P355NH steel in the joint zone was partially recrystallized through annealing. The normalizing caused not only the recrystallization of both materials, but also the formation of a diffusion zone and precipitates in Inconel 625. The clad steel plates are used in a variety of applications including the building of offshore structures, which may require repairs in water environment. Tomków et al. [24] found that the Temper Bead Welding effectiveness was experimentally verified as a method that may reduce the susceptibility to cold cracking in water environment. This provided a useful reference for the partial repair of clad steel plates in the underwater service.

However, there is little research on the heating process of metal composites. Before rolling of metal composite plate, heating temperature, heating time, and other parameters in the heating process directly affect the diffusion velocity of elements and interfacial microstructure, and then affect the comprehensive properties of metal composite plate. The combination of interface microstructure, element diffusion, mechanical properties, and possible defects is important for interface research, which has important guiding significance for the study of the processing technology of metal composite plates [25]. Therefore, this paper takes 2205 duplex stainless-steel (cladding material) with excellent corrosion resistance and Q235B carbon steel (base material) commonly used in the field of engineering stress as the research object to systematically study the influence of the heating process parameters on the interface characteristics and mechanical properties of 2205/Q235B composite plate.

## 2. Experimental Procedures

The cladding material and base material used in this paper are 2205 duplex stainless-steel and Q235B low carbon steel, respectively. The specific chemical composition is listed in Tables 1 and 2, and the size is 50 mm × 30 mm × 10 mm and 50 mm × 30 mm × 8 mm, respectively. The surfaces of two kinds of steel plates were ground using 1200# sandpaper by grinding with a grinder, and then cleaning the surface with alcohol to remove organic matter. Vacuum electron beam welding method was adopted to seal and weld the four sides of the two steel plates. The vacuum degree was 3.6 × $10^{-2}$ Pa, the welding voltage was 70 kV, the welding current was 35 mA, and the welding speed was 600 mm/min.

**Table 1.** Chemical composition of 2205 stainless-steel (wt.%).

| C | Si | Mn | Cr | Ni | Mo | S | P | N | Fe |
|---|---|---|---|---|---|---|---|---|---|
| 0.038 | 0.67 | 1.49 | 22.31 | 6.31 | 3.04 | 0.005 | 0.037 | 0.18 | Bal. |

**Table 2.** Chemical composition of Q235B low-carbon steel (wt.%).

| C | Si | Mn | S | P | Fe |
|---|---|---|---|---|---|
| 0.177 | 0.147 | 0.348 | 0.014 | 0.014 | Bal. |

To study the influence of heating temperature on interfacial structure and element diffusion, 2205/Q235B composite blank prepared by vacuum electron beam sealing and welding process was placed in a high-temperature heat treatment furnace for heating treatment. The heating temperatures were 1100 °C, 1150 °C, and 1200 °C, respectively, and the heating rate was 10 °C/s. The samples were insulated for 1 h. After taking out the samples, they were cooled to room temperature. Sample numbers were denoted as 1#, 2#, and 3#.

To study the influence of heating time on the interface structure and performance of 2205/Q235B composite billet, the heating temperature was set at 1200 °C, the heating speed was 10 °C/s, and the insulation was 1 h, 2 h, 3 h, and 4 h, respectively. After removing the samples from the oven, they were placed in a water tank to be quenched to room temperature, and their numbers were recorded as 4#, 5#, 6#, and 7#, respectively. Water quenching was used to return the high-temperature structure of the composite plate to room temperature and observe the diffusion of elements in the insulation stage.

The Ø10 mm, height of 6.7 mm 2205 duplex stainless-steel columns and Ø10 mm, height of 10 mm mild steel Q235B cylinder were docked face to face. The Gleeble 3800 thermal simulation machine (Data Sciences International, Saint Paul, MN, USA) was used to clamp the hammer head and conduct the compression simulation experiment at temperatures of 1100 °C, 1150 °C, and 1200 °C. Other heating process conditions are shown in Table 3. Sample numbers were denoted as 8#, 9#, and 10#.

**Table 3.** Heating progression parameters.

| Number | Heating Temperature (°C) | Holding Time (h) | Cooling Mode | Rolling Parameter | Rolling Pass | | |
|---|---|---|---|---|---|---|---|
| | | | | | 0 | 1 | 2 |
| 1# | 1100 | | | | | | |
| 2# | 1150 | 1 | Air-cooling | - | | | |
| 3# | 1200 | | | | | | |
| 4# | | 1 | | | | | |
| 5# | 1200 | 2 | Water-cooling | - | | | |
| 6# | | 3 | | | | | |
| 7# | | 4 | | | | | |
| 8# | 1200 | | | Temperature (°C) | 1200 | 1080 | 930 |
| | | | | Thickness (mm) | 16.7 | 13.36 | 10.69 |
| | | | | Deformation (mm) | | 3.34 | 2.67 |
| 9# | 1150 | 3 | Air-cooling | Temperature (°C) | 1150 | 1080 | 930 |
| | | | | Thickness (mm) | 16.7 | 13.36 | 10.69 |
| | | | | Deformation (mm) | | 3.34 | 2.67 |
| 10# | 1100 | | | Temperature (°C) | 1100 | 1080 | 930 |
| | | | | Thickness (mm) | 16.7 | 13.36 | 10.69 |
| | | | | Deformation (mm) | | 3.34 | 2.67 |

The metallographic specimens of the 2205/Q235B composite plate were polished according to the metallographic treatment standard. The 2205 duplex stainless-steel is etched by 40% KOH solution at a constant voltage of 8 V for 10 s. Q235B low-carbon steel is etched by 4% alcohol nitrate solution. The microstructure was observed by a Zeiss Scope (Carl Zeiss, Oberkochen, Germamy). A1 metalloscope and Zeiss Sigma 500/VP field emission scanning electron microscope, and the content and distribution of alloying elements were detected and analyzed by EDS. According to the standard GB/ T 1954-2008 [26], the ferritic content in the duplex stainless-steel side of the 2205/Q235B composite steel plate was measured by the ferritic meter SP10a.

Shear strength of the 2205/Q235B duplex stainless-steel composite plate samples was tested according to standard GB/T 6396-2008 [27]. Due to the small size of the sample, it was difficult to test shear strength by conventional methods. Shear samples with a size of 10 mm × 5 mm × 5 mm were

cut from the sample by wire cutting. The 2205 stainless-steel side is fixed in a combined fixture with grooves of the same size, so that the Q235B carbon steel side is exposed just outside the fixture. The fastening bolt was rotated at the top of the fixture to clamp the shear sample. Then it was placed on the shear test base for the shear test. Shear strength test was carried out on a MTS-E45 microcomputer controlled electronic universal test machine at a shear rate of 0.4 mm/min. The microhardness of the composite plate was measured by an HV-1000A microhardness tester. The load was 25 g and the loading time was 10 s. Testing was performed from the interface to the left and right and hardness was measured every 20 microns.

Electrochemical corrosion properties of the 2205 stainless-steel surface under different rolling processes were tested by a CHI660E electrochemical detection system. Tests were conducted in accordance with ASTM G5 [28] and ASTM G59 [29] in aqueous solution of 3.5 wt.% NaCl at room temperature. Electrochemical corrosion tests were carried out with standard three electrodes, i.e., saturated mercury-reference electrode, platinum electrode, and 2205 duplex stainless-steel as the working electrode. To obtain a stable voltage, the sample was first immersed in 3.5 wt.% NaCl solution for 10 min, and then the electrochemical impedance spectrum was tested. The test frequency range was $10^5$–$10^{-2}$ Hz, and the sine wave signal amplitude was 10 mV. To compare the corrosion resistance at the joints of the 2205/Q235B duplex stainless-steel composite plates rolled at different heating temperatures, cold inserts were used to seal five surfaces of the samples, exposing only the cross section. After grinding and polishing, the treated samples were immersed in HCl solution of 5 mol/L for 24 h, and the corrosion morphology was observed by SEM.

## 3. Results and Discussions

### 3.1. Microstructure of 2205/Q235B Duplex Stainless-steel Clad Plate

#### 3.1.1. Microstructure at Different Heating Temperatures

Figure 1 shows the interfacial microstructure of the composite samples heated at 1100 °C, 1150 °C, and 1200 °C. The interface is clean and straight, and the interfacial gap is completely closed by elemental diffusion. It can be seen that the metallurgical bonding of the 2205/Q235B composite billet can be achieved by mutual diffusion of elements only with elevated temperatures. The metallographic structure of the sample has obvious differences in different regions, which can be roughly divided into four regions: Q235B carbon steel region, decarburizing region, carburizing region, and the 2205 duplex stainless-steel region. Decarburizing and carburizing zones are caused by the diffusion of elements during heating and air cooling. As the content of C, Cr, Fe, and other elements in carbon steel and stainless-steel is significantly different, it provides the original driving force for elemental diffusion. The Q235B base layer far from the binding interface is composed of ferrite + pearlite structures. In the decarburization zone, during the heating process, C atoms in Q235B near the interface diffuse across the binding boundary to the side of stainless-steel, resulting in a decrease of carbon content in the pearlite, transformation to ferrite tissue, and continuous growth with elevated temperature. Therefore, the decarbonization zone is only coarse ferrite tissue. The carburizing zone is the continuous enrichment of C atoms to the stainless-steel side, near the interface of the Cr atoms in 2205 across the concentration gradient with continuous diffusion to the carbon steel side. With the increase of austenite promoting element C content and the decrease of ferrite promoting element Cr content, the duplex structure began to transform into continuous thick austenite bands. The 2205 cladding layer far from the binding interface still presents a ferrite + austenite duplex structure because it is not affected by elemental diffusion. The width of the decarburizing zone, carburizing zone, and grain size increased with the increase of heating temperature.

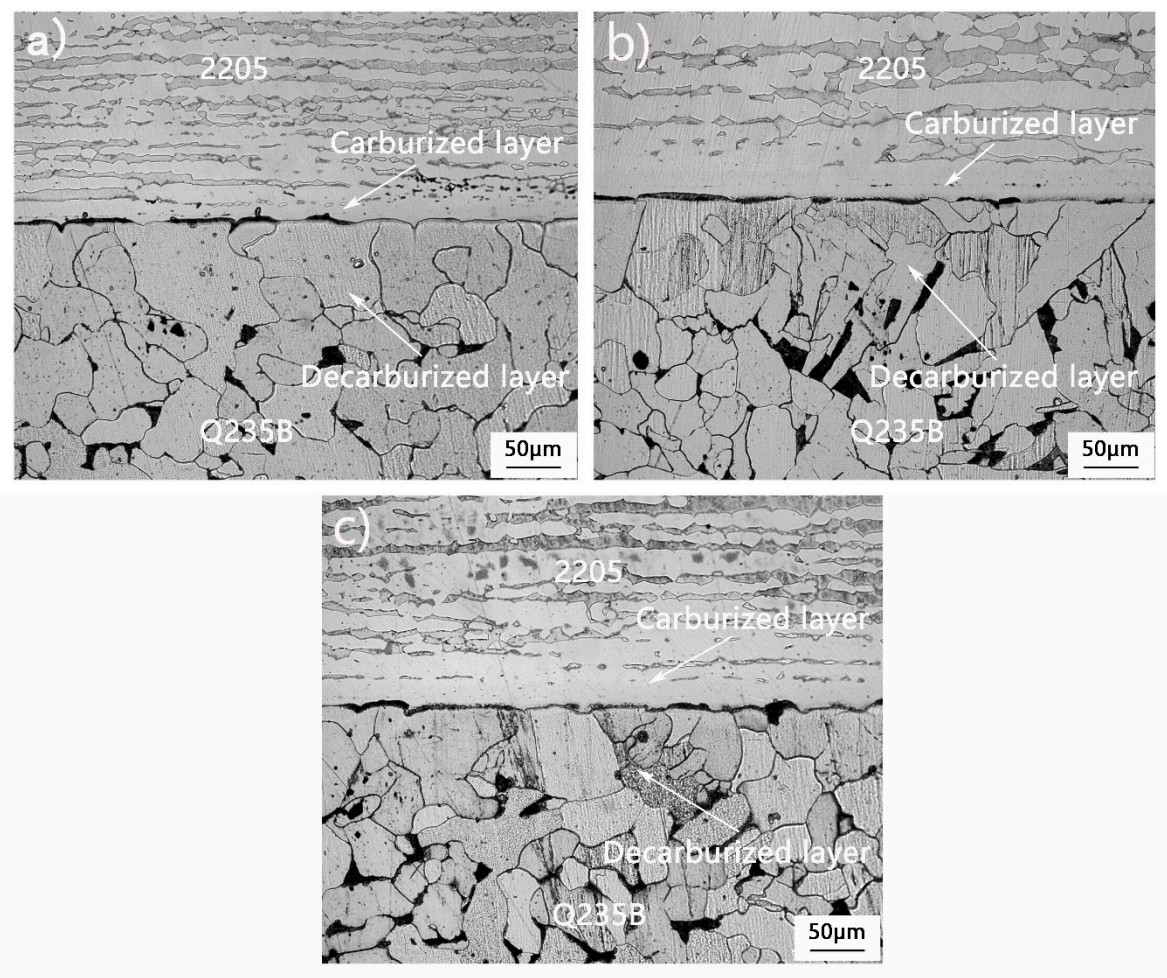

**Figure 1.** Microstructure of the cross section of a 2205/Q235B composite plate at different heating temperatures: (**a**) 1100 °C, (**b**) 1150 °C, and (**c**) 1200 °C.

Figure 2 shows the microstructure of 2205 stainless-steel away from the bonding interface. Heating in the range of 1100–1200 °C, the microstructure morphology of 2205 at room temperature was both ferrite and austenitic. The white protruding structure is austenite and the light-colored matrix is ferrite. In Figure 2a, the austenite structure has a small grain size consisting of small block austenite and slender austenite bars. As the heating temperature increases, the fine austenite grains grow and the slender austenite bar widths increase, transforming from the initial continuous structure to a more intermittent content, as shown in Figure 2b,c. Austenite is the non-equilibrium phase at high temperature. As heating temperature increases, the transformation of the non-equilibrium austenite into ferrite occurs, leading to higher ferrite content.

### 3.1.2. Microstructure under Different Heating Times

Figure 3 is the microstructure after water quenching at a holding time of 1–4 h at a temperature of 1200 °C. In the quenching process of rapid cooling, the thermal expansion coefficients of γ austenite and δ ferritic are different, so the volume contraction coordination is poor, and a large thermal stress is generated in the sample. When the thermal stress generated during quenching reaches the tensile limit of the tissue, crack nucleation will be induced during quenching, thus releasing the thermal stress. Austenite belongs to the hard phase with high strength, and its dynamic recovery capacity is significantly inhibited. Strain energy storage is not easy to be consumed through softening mechanism, thus inducing stress concentration in the austenite phase or at the interface of two phases. The ferrite phase is prone to dynamic recovery and can always maintain a low stress state. Compared with

austenite, ferrite shows better plastic deformation ability and can withstand larger strain. Therefore, under the action of quenching thermal stress, the sample cracked on the austenitic strip, as shown in Figure 3a. With the extension of heat preservation time, the organizational stress in the sample is fully released. In the subsequent quenching process, the stress level also decreases, and the quenching cracking tendency gradually weakens. When the insulation time reaches more than 2 h, quenching cracks will not appear in the sample, as shown in Figure 3b–d. In addition, when the insulation time is extended, the diffusion of C, Cr, and other elements occurs more readily, the diffusion distance continues to increase, and the width of the austenite band increases.

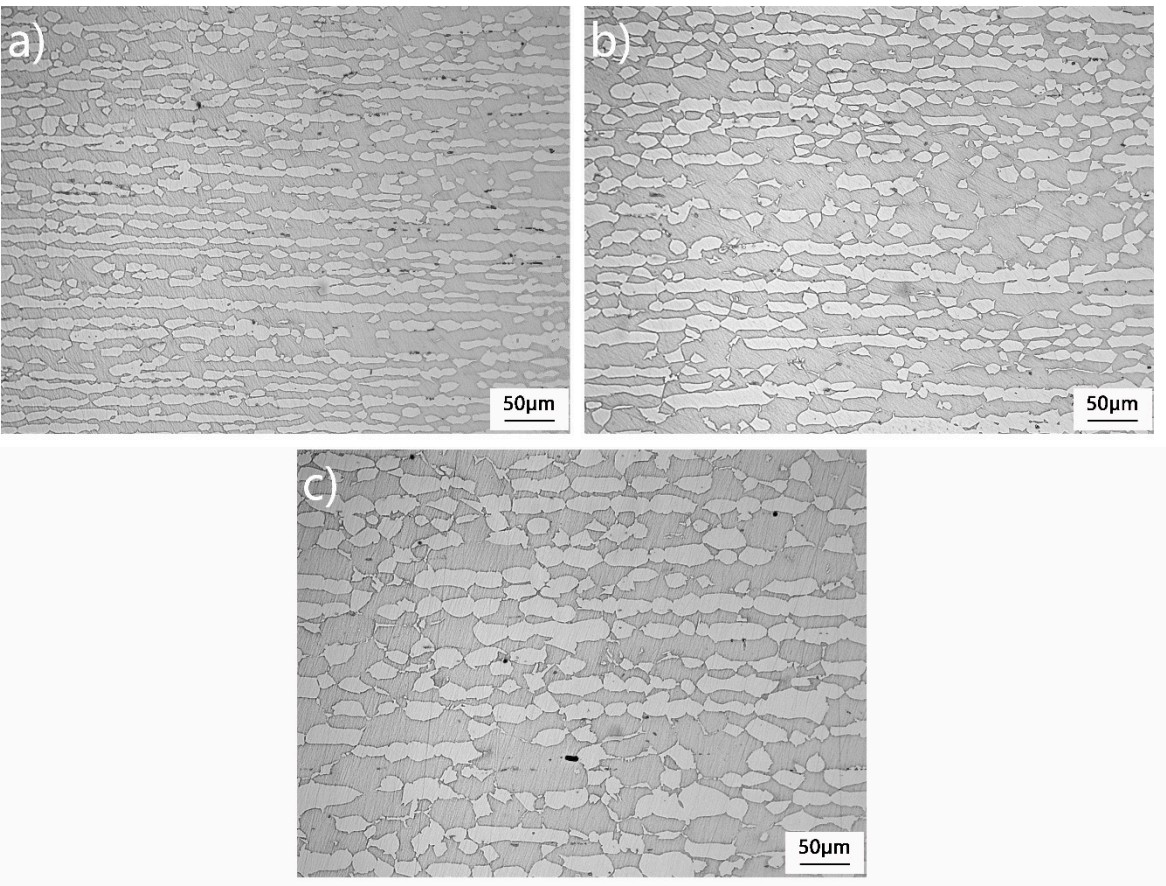

**Figure 2.** Microstructure of 2205 stainless-steel at different heating temperatures away from the bonding interface: (**a**) 1100 °C, (**b**) 1150 °C, and (**c**) 1200 °C.

When the heating temperature is the same, the effect of insulation time on the microstructure of 2205 duplex stainless-steel shows that the grain size increases with the extension of insulation time. Additionally, the prolonged insulation time is conducive to the diffusion of Cr, Mo, and other elements. The atomic density of the ferrite lattice is lower than that of the austenite lattice. The decrease in Cr, Mo, and other ferrite forming elements will lead to the transformation of ferrite into austenite [30,31], as shown in Figure 4a–d. In these Figures, Q235B belongs to low carbon steel. After heating and holding, rapid water cooling will generate slender strip martensite. In addition to lath martensite and ferrite distributed along grain boundaries, a small amount of feathery bainite also appeared in Figure 4e–h tissues. At the same time, due to the high quenching temperature, the lamellar martensite obtained after quenching is larger.

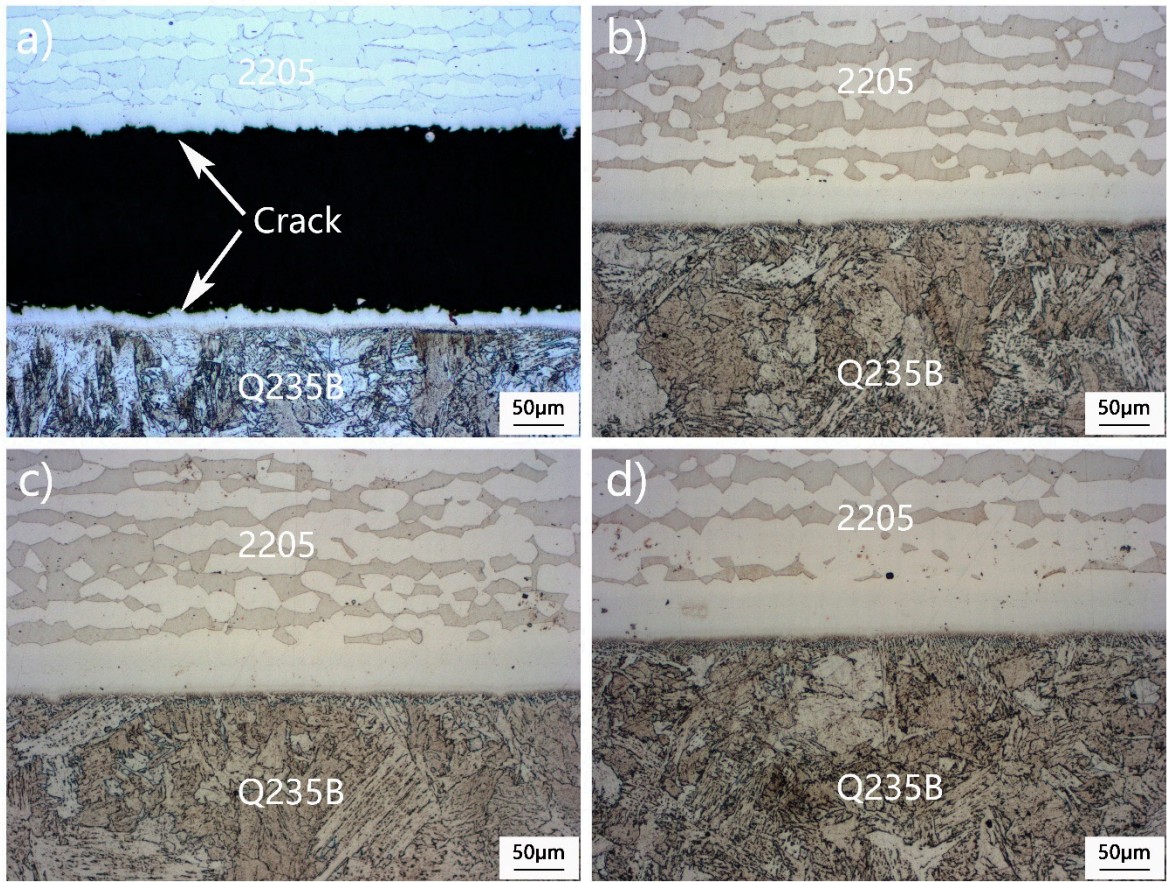

**Figure 3.** Microstructure of the cross section of the 2205/Q235B composite plate after different insulation times: (**a**) 1 h, (**b**) 2 h, (**c**) 3 h, and (**d**) 4 h.

### 3.1.3. Microstructure after Compression Deformation at Different Heating Temperatures

The 2205/Q235B composite plate was heated at 1100 °C, 1150 °C, and 1200 °C, respectively. The microstructure of the cross section after rolling under the same deformation conditions is shown in Figure 5. Compared with Figures 1 and 3, the grain size of 2205 stainless-steel and Q235B low carbon steel is refined, and the width of the decarburizing zone and carburizing zone near the bonding interface is also greatly reduced. The main reason for this is that during the compression deformation above 930 °C, the grain of the composite plate is constantly broken up, dynamically recovered and recrystallized, and the grain structure is refined. As the heating temperature increased, the width of the decarburization zone increased from 65.1 μm at 1100 °C to 79.4 μm at 1200 °C, as shown in Table 4. This is directly related to the diffusion of C atoms in carbon steel. As the heating temperature increases, the diffusion distance of C atoms is increased.

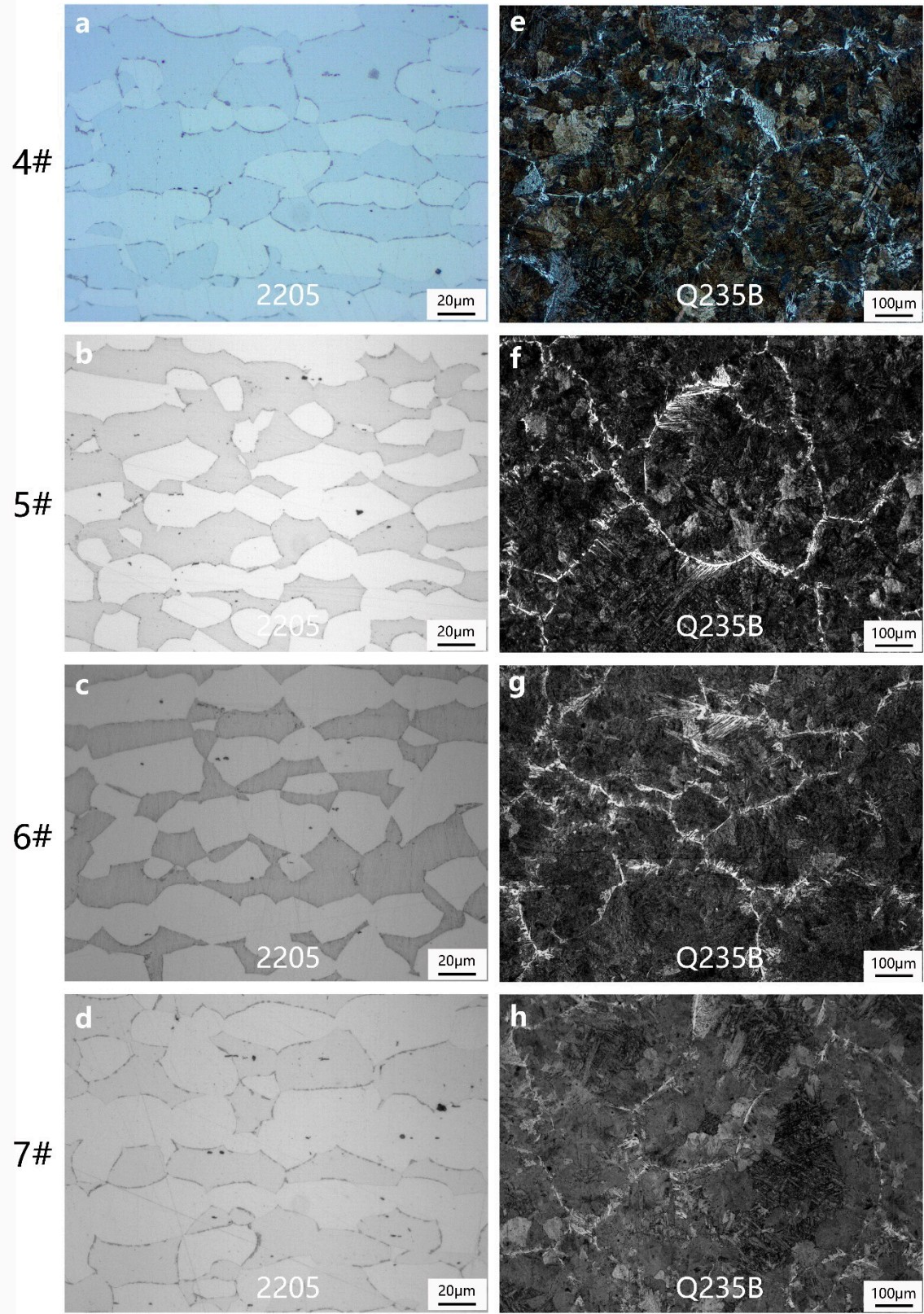

**Figure 4.** Microstructure of 2205 stainless-steel and Q235B low-carbon steel after different insulation times: (**a**,**e**)1 h, (**b**,**f**) 2 h, (**c**,**g**) 3 h, (**d**,**h**) 4 h.

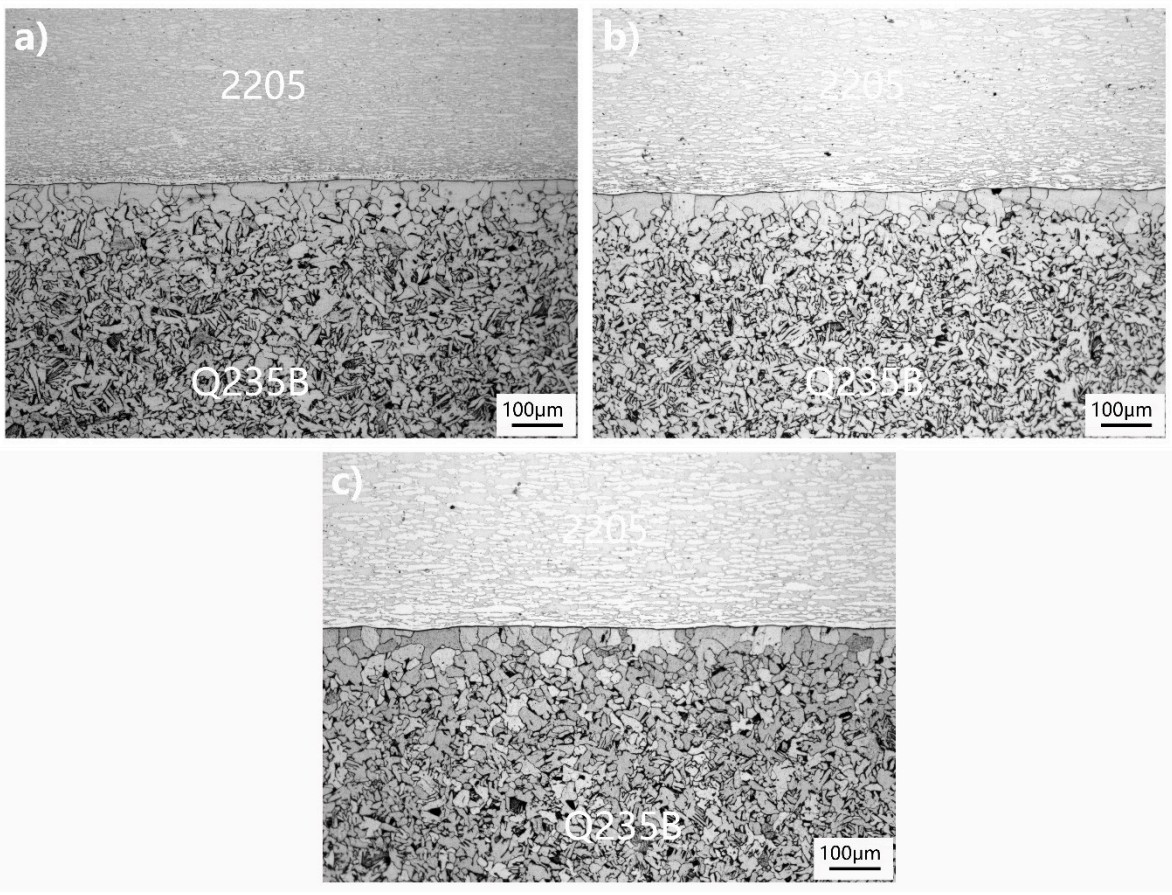

**Figure 5.** Microstructure of the cross section of a 2205/Q235B composite plate after rolling at different heating temperatures: (**a**) 1100 °C, (**b**) 1150 °C, and (**c**) 1200 °C.

**Table 4.** Thickness of Q235B low carbon steel (μm).

| Number | 1 | 2 | 3 | 4 | 5 | 6 | Average |
|---|---|---|---|---|---|---|---|
| 8# | 64.7 | 65 | 64.8 | 65.2 | 66.4 | 64.3 | 65.1 |
| 9# | 72.2 | 73.6 | 74.1 | 73.1 | 72.9 | 74 | 73.3 |
| 10# | 80.1 | 79.3 | 78.6 | 78.9 | 80.3 | 79.2 | 79.4 |

*3.2. Elemental Diffusion Analysis of the 2205/Q235B Duplex Stainless-steel Composite Plate*

Figure 6 is the detection result of the line element distribution at the interface of the 2205/Q235B composite plate. CPS stands for counts per second, which means the number of signals collected per second. From Tables 1 and 2, the content of C in Q235B carbon steel is 0.177%, the content of Fe is about 99%, and Cr, Ni, Mo, and other alloy elements are residual elements. The carbon content in 2205 stainless-steel is 0.038%. When Cr, Ni, Mo, and other alloy elements are added in large quantities, the content of Fe decreases significantly to approximately 66%. Near the interface, there will be a large concentration gradient of each element. The size of the C atom is smaller than that of Fe atom, and it can be diffused through the gap diffusion mechanism in the Fe lattice. The atomic sizes of Cr, Ni, and other atoms are similar to those of Fe, which diffuse via the vacancy diffusion mechanism. Since the diffusion activation energy of interstitial atoms is less than that of substitutional atoms, interstitial diffusion is more likely to occur than vacancy diffusion [32]. In theory, carbon atoms spread faster and farther than other atoms. However, as C is a light element, EDS detection cannot accurately reflect the

trend change in C content. Generally, it indirectly reflects the diffusion distance of the C atoms through the change of decarburization zone width at the composite interface. The equation [33] is defined as:

$$D = D_0 \exp(-\frac{Q}{RT}) \tag{1}$$

where $D_0$ as the diffusion coefficient, $Q$ as the diffusion activation energy, R for the gas constant, and $T$ is the thermodynamic temperature. It can be seen that temperature is the main variable affecting the diffusion coefficient, and the diffusion coefficient D increases sharply with the increase of temperature.

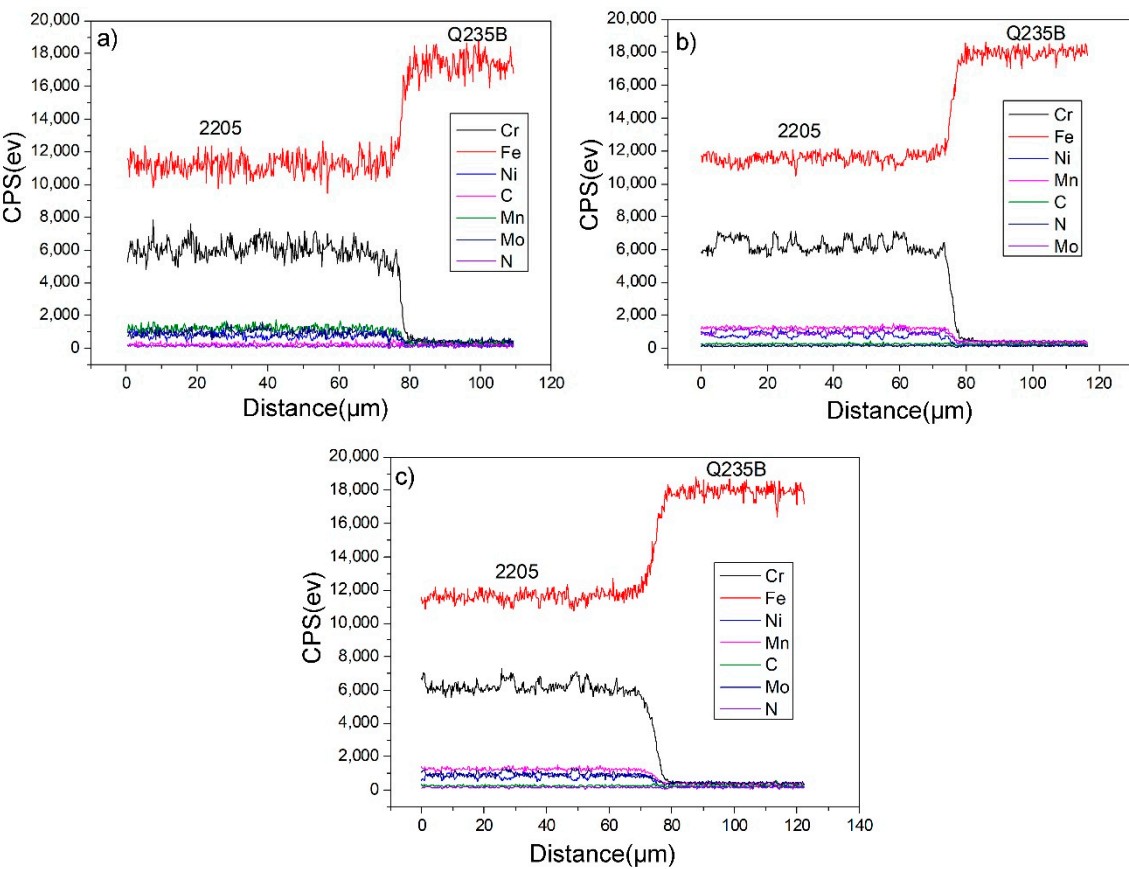

**Figure 6.** EDS test results of the 2205/Q235B composite steel plate binding zone: (**a**) 1100 °C, (**b**) 1150 °C, and (**c**) 1200 °C.

Figure 7 investigates chromium as the detection object and compares the diffusion law of Cr under three heating conditions. Under the same insulation time, as the heating temperature gradually increases, the slope of the diffusion region curve decreases, and the diffusion distance of Cr gradually increases from 18.2 μm to 41.2 μm, as shown in Figure 7a. At the same heating temperature (Figure 7b), the diffusion distance of Cr gradually increases as the insulation time increases. When it was insulated for 2 h and 3 h, the curves had similar slopes, i.e., the diffusion distance was similar at 52.8 μm and 56.1 μm, respectively. When the insulation time increased to 4 h, the slope of the curve decreased significantly, and the diffusion distance increased significantly to 77.1 μm. As shown in Figure 7c, the curve is steep after compression deformation, and the diffusion distance of Cr is significantly reduced, ranging from 4.1 to 10.2 μm. In addition, the Cr content curve of 2205 stainless-steel side in Figure 7 fluctuates up and down. This is because the ferrite content in 2205 duplex stainless-steel tissues is slightly higher than austenite and when scanned to the ferrite area, Cr content significantly increased, and when scanning to the austenite area, Cr content began to decrease.

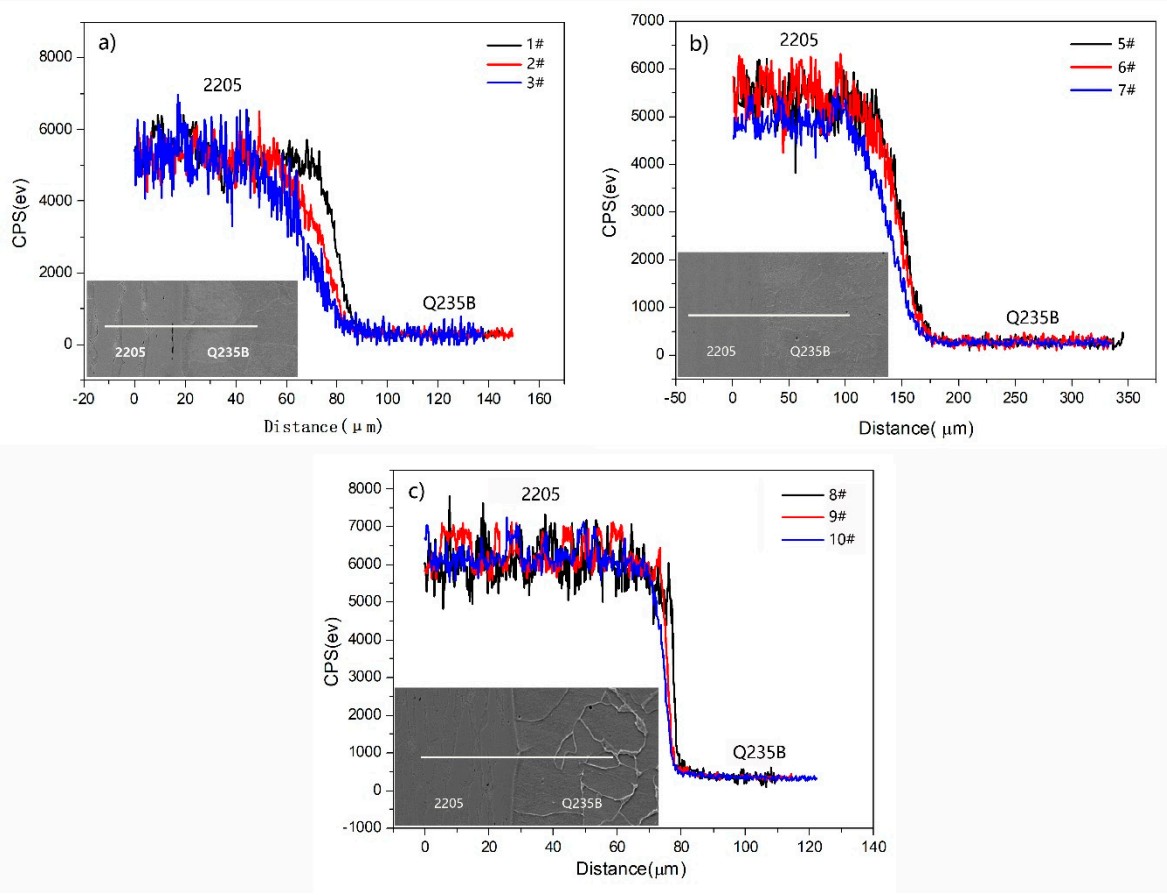

**Figure 7.** Distribution curve of Cr in the bonding zone of 2205/Q235B composite steel plate: (**a**) 1#, 2#, and 3#, (**b**) 5#, 6#, and 7#, (**c**) 8#, 9#, and 10#.

### 3.3. Shear Strength of 2205/Q235B Duplex Stainless-steel Composite Plate

The shear strength of the 2205/Q235B duplex stainless-steel composite plate under different heating conditions is shown in Figure 8. Samples 1#, 2#, and 3# did not undergo compression deformation, and the cooling speed was relatively slow in the air-cooling process. The samples stayed in the high temperature stage for a long time, with large grain size and shear strength between 114 MPa and 132 MPa, as shown in Figure 8 (a). It fails to meet the shear strength standard (210 MPa) specified in GB/T 8165-2008 [34] and cannot meet the requirements of engineering applications. The recrystallization grain sizes of samples 8#, 9#, and 10# are primarily affected by deformation temperature under conditions of constant deformation and final rolling temperature. As the initial rolling temperature increases, the size of the recrystallization region expands. In addition, the decarburization zone is a weak area near the composite interface. With the increase of the initial rolling temperature, the diffusion of C is more complete and the ferrite structure is relatively large. Under the joint action of the two, the shear resistance of the samples decreases slightly with the increase of the initial rolling temperature, as shown in Figure 8 (b). After 1 h of insulation at 1200 °C, the sample was quenched with water, resulting in cracking at the composite interface, as shown in Figure 3a. When the insulation time of the sample is more than 2 h, the sample will not crack after water quenching, and the shear strength is 329 MPa, 322 MPa, and 375 MPa, respectively, as shown in Figure 8 (c). The shear strength of the samples is higher than that of the rolling deformation samples, because the decarburization structure of the samples changes to a martensite structure after water quenching, which improves the shear strength of the samples.

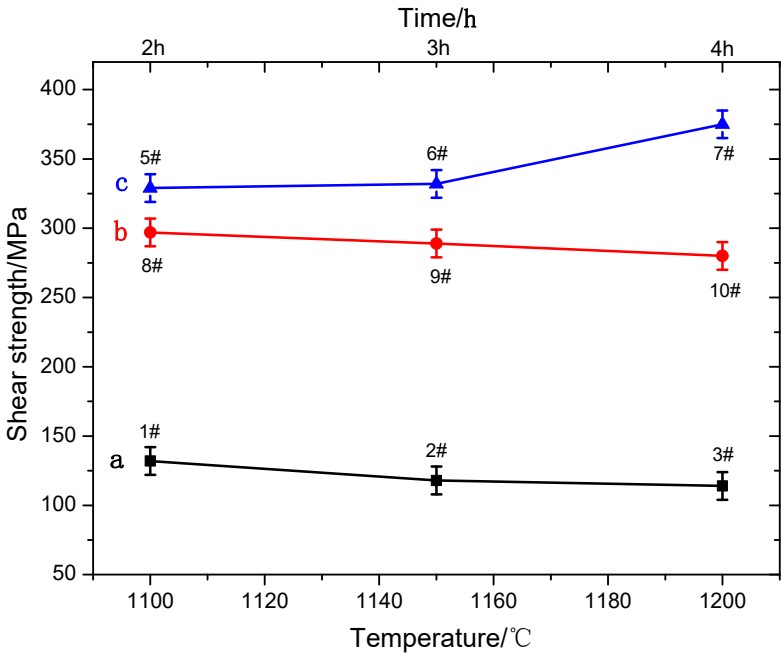

**Figure 8.** Shear strength of the 2205/Q235B composite steel plate.

### 3.4. Microhardness of the 2205/Q235B Duplex Stainless-steel Composite Plate

The effects of temperature and time on the composite process were investigated in samples 1#–3# and 5#–7#. It was found that the microstructure and grain size of the samples were significantly different from that of the samples after the rolling composite. Meanwhile, to further analyze the influence law of different heating temperatures on the composite effect under the same deformation condition, samples of 8#, 9#, and 10# were selected to test and analyze the microhardness of the bonding surface and nearby area, as shown in Figure 9. The microhardness at the interface of the sample is roughly 236–256 HV, higher than that of Q235B and lower than that of 2205. From the joint interface, the microhardness of the carbon steel decreased significantly and gradually leveled off after about 60 μm. This is roughly consistent with the decarbonization zone width between 65.1 μm and 79.4 μm in Table 4. This is because the C element on the side of Q235B diffuses to form the decarburization zone, and the structure is mainly rough ferrite soft phase, resulting in a sharp decrease in hardness. As it moves away from the binding interface and enters the normal region of ferrite + pearlite of Q235B, the microhardness tends to be stable, roughly within the range of 170–184 HV, which conforms to the normal hardness value of Q235B. The microhardness increased gradually from the bonding interface to 2205 side and leveled off 20 μm later. This is because the diffusion distance of Cr is between 4.1 μm and 10.2 μm, so the microhardness of the sample beyond 20 μm from the bonding interface is the microhardness of 2205 base material. With the increase in rolling temperature, the microhardness of sample 2205 of 8#, 9#, and 10# gradually decreases. The analysis shows that with the increase of rolling temperature, austenite changes to ferrite and the content of soft ferrite gradually increases (as seen in Figure 11 and Table 5), so that the hardness of the 2205 cladding layer decreases gradually.

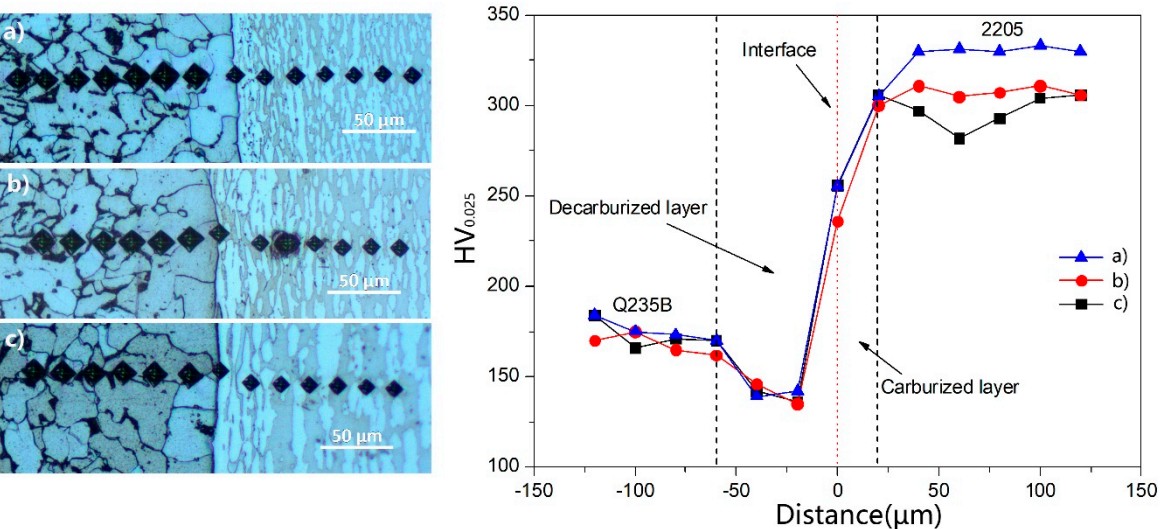

**Figure 9.** Microhardness of the 2205/Q235B clad sheet steel: (**a**) 8#, 1100 °C, (**b**) 9#, 1150 °C, and (**c**) 10#, 1200 °C.

**Table 5.** Ferrite content on the surface of the 2205 duplex stainless-steel (%).

| Number | 1 | 2 | 3 | 4 | 5 | Average |
|--------|------|------|------|------|------|---------|
| 8# | 45.9 | 44.3 | 45.9 | 43.7 | 45.9 | 44.9 |
| 9# | 47.4 | 47.6 | 47 | 47.6 | 47.7 | 47.5 |
| 10# | 49.7 | 50.5 | 49.1 | 49.6 | 49.7 | 49.6 |

*3.5. Corrosion Resistance of 2205/Q235B Duplex Stainless-steel Composite Plate*

Three samples (8# 9#, and 10#) were selected to test the electrochemical impedance of the cladding surface at different heating temperatures, as shown in Figure 10. According to the principle of electrochemical impedance measurement, the diameter of the curves in the Nyquist plots reflects the impedance [35]. Thus, it can be seen that the curve diameter of the deformation sample heated at 1200 °C is the largest, that is, the impedance value is the largest, while The curve diameter of the sample heated at 1150 °C was the second and the impedance value was in the middle, and finally, the curve diameter and impedance of the compressed samples were the smallest when heated at 1100 °C. Therefore, the corrosion resistance of the samples were the greatest and the least when heated at 1200 °C and 1100 °C, respectively.

The corrosion resistance of 2205 duplex stainless-steel is closely related to δ ferrite content, and the δ ferrite and γ auxin ratio is close to 1:1, where the corrosion resistance of the material is the best. Ferrite content in 2205 duplex stainless-steel of 2205/Q235B composite plate is detected by a ferrite meter, and the results are shown in Table 5. From Figure 11, as temperature increases, the more beneficial the transformation of γ austenite to δ ferrite, thus increasing the content of δ ferrite. When the heating temperature reached 1200 °C, the sample with a δ and γ ratio of 1:1, the corrosion properties of the material were the best, which corresponds with the test results of electrochemical impedance spectroscopy.

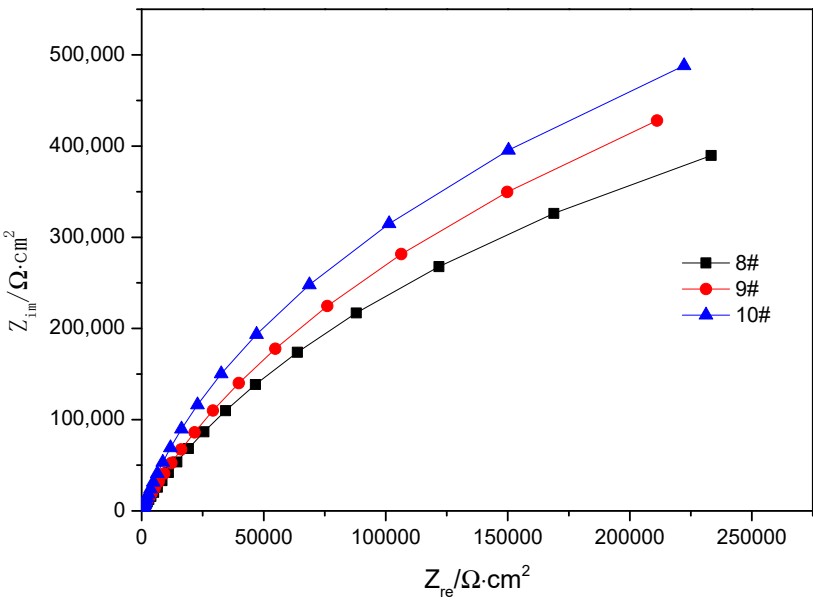

**Figure 10.** Surface impedance mapping on the surface of 2205 duplex stainless-steel.

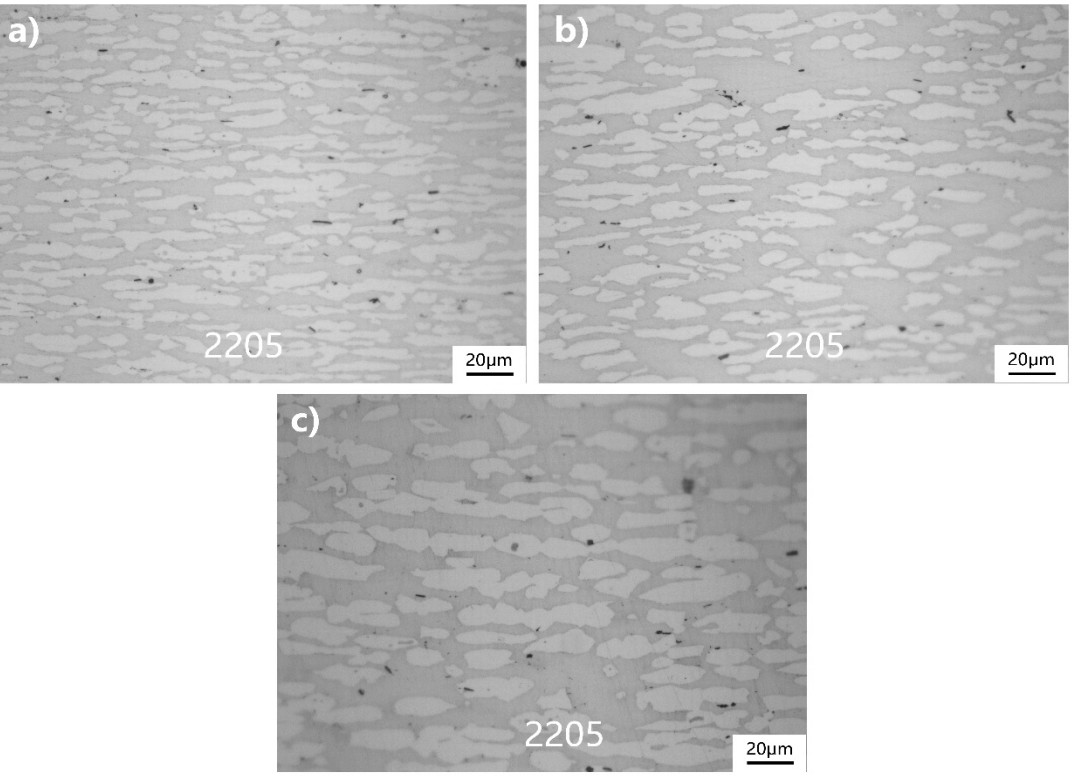

**Figure 11.** Ferrite distribution on the surface of 2205 duplex stainless-steel: (**a**) 1100 °C, (**b**) 1150 °C, and (**c**) 1200 °C.

The corrosion resistance of 2205 duplex stainless-steel is closely related to the δ ferritic content. The δ ferrite to γ austenite ratio is close to 1:1 when the corrosion resistance of the material is the best. Ferrite content in 2205 duplex stainless-steel of 2205/Q235B composite plate is detected by ferrite meter, and the results are shown in Table 5. It can be seen from Figure 11 that the higher the heating temperature, the more beneficial the transformation of γ austenite to δ ferrite, thus increasing the content of δ ferrite. When the heating temperature reaches 1200 °C, the ratio of δ ferrite to γ austenite

in the sample trends towards 1:1, and the corrosion properties of the material are the best. This is consistent with the results of electrochemical impedance spectroscopy.

Figure 12 shows the micromorphology of the cross section of the 2205/Q235B composite plate soaked in 5 mol/L HCl solution for 24 h. By comparison, it was found that the stainless-steel side of the three groups of 2205/Q235B composite plates did not change significantly, and only the grain boundary was corroded, while the side of the Q235B was seriously corroded, and the corrosion was the most significant near the bonding area. With the increase in heating temperature, the width of the corrosion pit near the bonding zone on the side of Q235B low carbon steel increases gradually. The analysis shows that: (1) In corrosive medium, the impedance value of 2205 duplex stainless-steel is much higher than that of Q235B low carbon steel, so low carbon steel corrosion is more significant, and (2) the carbon poor zone is formed near the bonding zone of Q235B low carbon steel, which is mainly ferrite. Potential difference is formed not only with other areas of Q235B low carbon steel, but also with 2205 duplex stainless-steel. Therefore, this area is the most corroded. Under the action of stress, the corrosion speed of this area will be accelerated, and it is easy to form cracking in the form of the crack source. Therefore, in the application of 2205/Q235B duplex stainless-steel composite plate, the corrosion protection of the exposed section should be complete.

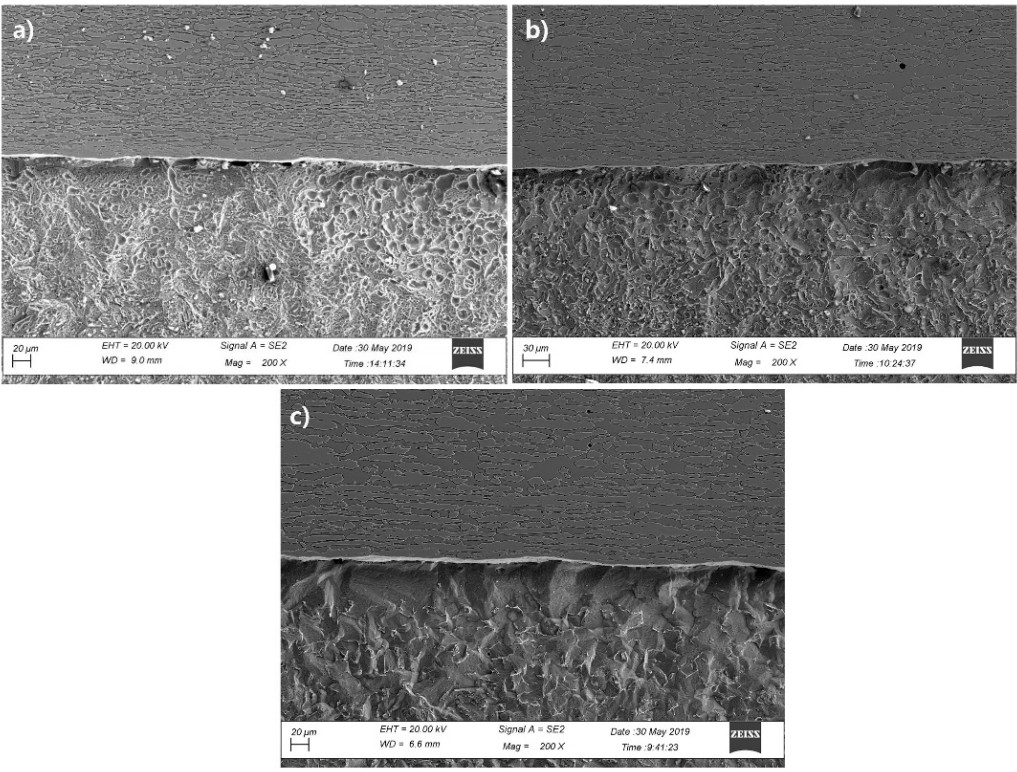

**Figure 12.** Acid immersion micromorphology of the cross section of the 2205/Q235B composite steel plate: (**a**) 1100 °C, (**b**) 1150 °C, and (**c**) 1200 °C.

## 4. Conclusions

(1) The interface of the 2205/Q235B composite billet realized metallurgical bonding by mutual diffusion of elements, and the ferrite content on the side of 2205 increased gradually with the increase in heating temperature.

(2) The microhardness of the decarburization zone in the rolling deformation sample was the lowest, and the microhardness at the bonding interface was about 236–256 HV. Under the same rolling deformation conditions, the width of the decarburization zone structure increased with the increase of heating temperature.

(3)     Under the same insulation time, the diffusion distance of Cr gradually increased with the increase of heating temperature. At the same heating temperature, the diffusion distance of Cr gradually increased with the holding time. After rolling deformation, the diffusion distance of Cr was significantly reduced to 4.1–10.2 μm.

(4)     The air-cooled samples have coarse microstructure due to the slow cooling speed, and the shear strength was only 114–132 MPa, which was significantly lower than the water-quenched samples and the rolling deformation samples. It cannot meet the requirements of engineering applications.

(5)     The corrosion rate of the rolled deformation specimens was the minimum when the coating was heated at 1200 °C. After the cross-section immersion test of 2205/Q235B composite plate, the width of the corrosion pit near the bonding zone on the side of Q235B low carbon steel was gradually increased with the increase in heating temperature.

**Author Contributions:** Conceptualization, F.X. and D.W.; methodology, F.X.; formal analysis, F.X. and Z.G.; investigation, F.X.; resources, L.Z.; writing—original draft preparation, F.X.; writing—review and editing, D.W. and Z.G.; supervision, D.W.; project administration, Z.G. and L.Z.; funding acquisition, D.W., F.X., Z.G. and L.Z.

**Acknowledgments:** This research was funded by Shandong Taishan Industry Leading Talents Project, grant number SF1503302301 and The APC was funded by Shandong Taishan Industry Leading Talents Project.

**Conflicts of Interest:** The authors declare no conflict of interest.

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
