# Peer review of "Effect of Heating Process on Microstructure and Properties of 2205/Q235B Composite Interface"

_metals, doi:10.3390/met9101027_

Round 1

Reviewer 1 Report

Dear Authors,
I have reviewed your article entitled "Effect of heating process on microstructure and properties of 2205/Q235B composite interface".
I have some suggestions, that I listed below.
General remarks
- I propose to add some references that were published in last 3 years. Now, there are only a few position, which the most is in Chinese.
- You should add some references from Metals. This journal includes very valuable articles that may be helpful to improve your paper.
- Please mark a), b), c)... in your figures.

Introduction
- You have presented the background to your researches and for me it looks good in this article, however as I wrote, you should add some new references.

I have one remarks, that could be helpful for your further investigations.
In line 36, you have wrote, that metal composite materials could be used in marine engineering. In line 50-52 you have added that "for further improve the comprehensive properties of these materials, researchers have conducted considerable research on heat treatment technology." Marine constructions could work in the water and may undergo failure. In this environment the traditional heat treatment is almost impossible. However, there is one repairing method called "temper bead welding" that used local heat treatment in the water and for carbon steels it provides to positive changes in microstructure. In last two years some results heve been published in this field.

Experimental procedures

- You have presented the chemical composition of used materials. Have ou prepared analysis of its composition or you presented values from standards?
- How many specimens have you prepared? How many were tested in each test?
- Lines 92-94 - why these parameters? From previous investigations, from references?
- Lines 105-106 - the cross-sections were polishing.

Results and discussion
- In my opinion this section is the strongest part of the reviewed manuscript. In my opinion the Authors have prepared discussion in high level. It is strongly connected with obtained results. I have small questions:
- line 241 - please add references to equation.
- Figures 6 and 7- please name "CPS", it should be clear for potential readers.
- There are couple grammatical mistakes in this section. Pleas check it.

Reviewer 2 Report

This is a very interesting work that fits well within the scope of this Journal. However, some aspects need to be addressed prior to publication of this article. Minor revisions are due.

Table 3: how many specimens were made for each group?

Table 3: the authors should rename the group so it is more simply for the readers to understand the figure. If the authors change the name of the groups, they have to change these names also in all figures.

Fig. 6. It is impossible to read the legend and the size of figure is small. Please increase it.

Fig. 9. On the ordinate axis the author should change “HV” with “HV0.025”.

Modarate proof of English is due.

Reviewer 3 Report

Dear authors,

thank you for the interesting work, but i have some comments.

line 30 and following: please check the terms of composite technologies, maybe the processes should be addressed as cladding technology.

line 104: Table 3. adjust column for Group

lines 106 to 110: change "corroded" into "etched"

line 117: please change matellographic images into something more common

Please divide the experimental results and the discussion and general findings into minimum two separate subsections.

Please improve the conclusion, and the article, with some general findings, in the present state it only summarizes the experiments.

Regards,

A Reviewer

Round 2

Reviewer 3 Report

Dear Authors,

thank You very much for the changes made. 

I understand the intention behind your statement, and can 

accept it. 

Regards 

a reviewer